# Trends in the Development of Antibody-Drug Conjugates for Cancer Therapy

**DOI:** 10.3390/antib12040072

**Published:** 2023-11-03

**Authors:** Chi Hun Song, Minchan Jeong, Hyukmin In, Ji Hoe Kim, Chih-Wei Lin, Kyung Ho Han

**Affiliations:** 1Department of Biological Sciences and Biotechnology, Hannam University, Daejeon 34054, Republic of Korea; chihun312@gmail.com (C.H.S.); alscks050@naver.com (M.J.); gur0793@naver.com (H.I.); hera42634@gmail.com (J.H.K.); 2Institute of Biochemistry and Molecular Biology, China Medical University, Taichung 406, Taiwan; cwlin25@mail.cmu.edu.tw

**Keywords:** antibody drug conjugates, monoclonal antibody, payload, linker, conjugation

## Abstract

In cancer treatment, the first-generation, cytotoxic drugs, though effective against cancer cells, also harmed healthy ones. The second-generation targeted cancer cells precisely to inhibit their growth. Enter the third-generation, consisting of immuno-oncology drugs, designed to combat drug resistance and bolster the immune system’s defenses. These advanced therapies operate by obstructing the uncontrolled growth and spread of cancer cells through the body, ultimately eliminating them effectively. Within the arsenal of cancer treatment, monoclonal antibodies offer several advantages, including inducing cancer cell apoptosis, precise targeting, prolonged presence in the body, and minimal side effects. A recent development in cancer therapy is Antibody-Drug Conjugates (ADCs), initially developed in the mid-20th century. The second generation of ADCs addressed this issue through innovative antibody modification techniques, such as DAR regulation, amino acid substitutions, incorporation of non-natural amino acids, and enzymatic drug attachment. Currently, a third generation of ADCs is in development. This study presents an overview of 12 available ADCs, reviews 71 recent research papers, and analyzes 128 clinical trial reports. The overarching objective is to gain insights into the prevailing trends in ADC research and development, with a particular focus on emerging frontiers like potential targets, linkers, and drug payloads within the realm of cancer treatment.

## 1. Introduction

Biopharmaceuticals encompass medications produced from substances sourced from living organisms, comprising recombinant proteins, biosimilars, and BioBetter drugs. These therapeutic agents are characterized by their minimal toxicity and reduced side effects, delivering remarkable efficacy in the precise treatment of specific diseases [1].

Traditional chemotherapy agents, known as first-generation treatments, have been employed in cancer therapy for a long time. In contrast, second-generation targeted anticancer drugs have emerged as a rapidly expanding category of therapeutic options, boasting remarkable specificity in inhibiting the growth and progression of cancer cells. These targeted anticancer drugs predominantly fall into two principal categories. The first group consists of tyrosine kinase inhibitors, which function by obstructing critical cell signaling proteins responsible for processes like cell proliferation and migration. Tyrosine kinase inhibitors gained substantial recognition in the realm of anticancer medications, notably highlighted by the success of Imatinib, a drug used in the treatment of chronic myeloid leukemia [2]. The second category encompasses monoclonal antibodies, which effectively obstruct the binding of receptors on the cell surface. Employing monoclonal antibodies in anticancer treatments triggers the specific initiation of cancer cell apoptosis and results in fewer side effects compared to conventional chemotherapy agents.

Third-generation immuno-oncology drugs encompass a range of approaches, including adaptive cell therapies, immune checkpoint inhibitors, and cancer vaccines, all designed to specifically recognize cancer cells and activate a patient’s immune system for the purpose of eliminating cancer cells [1,3].

Utilizing monoclonal antibodies in cancer therapy offers several benefits, including the selective induction of cancer cell apoptosis, precise targeting, extended duration of action within the body, and minimal side effects. For instance, Rituximab specifically targets CD20 on the surface of B lymphocytes, and depletes these cells through mechanisms such as antibody-dependent cellular cytotoxicity (ADCC), complement-dependent cytotoxicity (CDC), and apoptosis. Additionally, there have been developments in antibody drug conjugates (ADCs), which combine these monoclonal antibodies with cytotoxic drugs to augment the potency of targeted anticancer treatments [4].

ADCs represents a cutting-edge anticancer drug with a specific mechanism of inducing cell death in cancer cells. This drug comprises three essential components: an antibody, a payload, and a linker. Its structure entails an antibody that binds to the target substance’s epitope, a payload responsible for causing cell death, and a linker that connects the antibody to the payload. ADCs offer a significant advantage due to their monoclonal antibodies ability to selectively target and act on cancer cells, thereby reducing non-specific toxicity to healthy cells. Linkers can be categorized as cleavable and non-cleavable. Cleavable linkers release payloads into cancer cells, triggering cell death by exploiting the pH difference between cancer cells and normal cells. Non-cleavable linkers, although their exact mechanism remains unknown, release payloads through enzymatic degradation of lysosomes in targeted cancer cells. There are primarily two types of payloads in use: DNA-targeted and microtubule-targeted payloads. DNA-targeted payloads bind to the DNA of cancer cells, disrupting its structure and leading to cell death. On the other hand, microtubule-targeted payloads disrupt microtubules within cancer cells, resulting in cell cycle arrest and apoptosis. Creating an ADCs involves the conjugation of an antibody, linker, and payload. This can be achieved by utilizing amino acids present on the antibody’s surface or through artificial modification of the antibody surface. Two critical factors influencing the stability of an ADCs are the number of payload drugs conjugated to the antibody and the conjugation site of the antibody and linker. The modulation of these factors can be categorized into stochastic conjugation and site-specific conjugation methods [5].

The inception of the first generation of ADCs dates back to the mid-20th century, but they grappled with limitations such as inadequate control over drug binding sites on antibodies, reduced efficiency, and undesirable side effects. To address these issues, the second generation of ADCs employed antibody modification techniques, enabling better management of the drug-antibody ratio (DAR), amino acid substitution with payloads, integration of unnatural amino acids, and the selective attachment of drugs to specific antibody sites through enzymatic processes. In more recent advancements, the third-generation ADCs have introduced a technology that enables the steady attachment of drugs to antibodies without impeding the interaction between antigens and antibodies. Notably, among domestic companies, Pinobio’s NTX-301 stands out as a representative example of these innovative developments [6]. The pharmaceutical market comprises both synthetic drugs and biopharmaceuticals, with the biopharmaceuticals experiencing rapid growth. In 2017, biopharmaceuticals account for eight of the world’s top-selling medications. According to an IQVIA report, anti-cancer drugs are projected to become the most consumed category in the global drug market by 2025. The anti-cancer drug market is anticipated to expand by approximately 9–12% from 2021 to 2025, reaching a market size of $266.2 billion. As of 2021, the domestic drug market stood at $19.14 billion, ranking 13th globally and representing 1.5% of the global market. In terms of disease-specific growth, anti-cancer drugs and immunotherapy drugs are expected to see the most significant developments over the next five years. In the realm of anti-cancer drugs, including immuno-oncology and gene therapy medications, more than 100 new drugs are slated for launch in the next five years. The market is poised for rapid growth, projected to increase by 13–16% over five years from its 2022 valuation of $187.9 billion [7]. The ADC market, valued at $5.957 billion in 2020, is expected to expand from $6.8 billion to $15.4 billion between 2024 and 2029. Recent trends show a rise in the number of ADC candidates, owing to collaborative research and development efforts between global pharmaceutical companies and small to medium-sized biotechnology firms. Consequently, the market share of ADC is predicted to keep ascending [8]. Since the year 2000, the U.S. FDA has approved and introduced 12 ADCs into the market. Notably, nine of these approvals have occurred since 2017, with Zynlonta^®^’s loncastuximab tesirine-lpyl and Tivdak^®^’s tisotumab vedotin-tftv receiving approval in 2021. The most recent ADC to secure approval was ELAHERE^®^’s mirvetuximab soravtansine in the year 22 [7,9,10].

In this study, we analyzed a total of 12 ADCs available in the market. Furthermore, we reviewed recent publications from 71 related to ADC research and development. Additionally, we conducted a review of 128 clinical trial reports that encompassed clinical targets, diseases, linkers, payloads, conjugates, mechanisms of action, and clinical trends. Through this comprehensive analysis, we aim to elucidate the current research and development trends in ADC and their potential for fostering advancements in the field of anticancer therapeutics.

## 2. ADC Structure

ADC possesses a tripartite structure, comprising an antibody designed for precise antigen binding, a potent and cytotoxic drug payload capable of triggering cell apoptosis, and a linker facilitating the connection between the antibody and payload (Figure 1). To transform ADC into a viable therapeutic option, three pivotal mechanisms of action come into play. Firstly, the antibody must exhibit a specific and selective affinity for an antigen or receptor present on the target cell’s surface. Secondly, the payload needs to successfully internalize within the target cell and subsequently engage with the designated target to initiate programmed cell death. Lastly, throughout the process of antibody and payload binding, as well as their internalization within the target cell, the linker must efficiently dissociate the payload at the appropriate moment to ensure a stable bond between the antibody and payload is maintained [11]. In the context of an ADC featuring all three essential components, a monoclonal antibody exhibits the capacity to selectively attach to and enter cancer cells characterized by an overexpression of antigens. Subsequently, it undergoes a series of intracellular processes, commencing with the formation of early endosomes, progressing to their maturation into late endosomes, and ultimately culminating in their association with lysosomes [12]. The ADC’s payload undergoes cleavage within lysosomes, either through chemical or enzymatic processes. Upon release, this payload exerts its cytotoxic effect by disrupting DNA or microtubules within the targeted cell, ultimately leading to cell death [13] (Figure 2). Moreover, if the released payload possesses the capability to permeate cell membranes, it can trigger a bystander effect, causing neighboring target cells to undergo apoptosis without requiring ADC internalization [14]. Consequently, owing to the precise antibody binding and the pro-apoptotic nature of the payload, ADC offers several advantages, including a high therapeutic efficacy, cancer cell-specific targeting, relatively lower toxicity towards normal cells, and diminished side effects compared to conventional chemical cancer treatments [15].

### 2.1. Antibody

Antibodies are glycoproteins that exhibit a remarkable ability to selectively bind to specific antigens, eliciting a potent immune response. In order to harness their potential for ADCs applications, certain key criteria must be met, including efficient cellular internalization, minimal immunogenicity, and an extended plasma half-life. Typically, these antibodies molecules take the form of monoclonal antibodies designed to target a single antigen [16,17,18]. The choice of Antibodies for ADCs development hinges on diverse internalization mechanisms, contingent upon the expression of the target antigen within the recipient cell. The number of antigens found on each cell varies according to cancer type, disease stage, and individual patient traits. In general, antigen copy numbers can span from several thousand to tens of thousands per cell. For example, in breast cancer cases positive for HER2, antigen copy numbers can range from 100,000 to 2 million per cell. Similarly, in acute myeloid leukemia (AML) blasts, CD33 copy numbers may range from 1000 to 10,000 per cell [19,20,21]. Therefore, careful selection of an appropriate target antigen is imperative to ensure effective penetration of ADC into the cell. Equally crucial is the need for the target antigen to be primarily expressed in the target cell, while exhibiting lower levels in normal cells.

The promise of an ADC for therapeutic applications hinges significantly on the expression levels of the target antigen on cancer cells. An ideal target antigen profile for an ADC should encompass several key factors. High antigen expression ensures effective binding of a substantial number of ADC molecules to cancer cells, promoting therapeutic efficacy. Uniform antigen expression across these cells reduces the potential for diverse responses, while minimal expression on healthy tissues mitigates off-target effects and associated side effects. Moreover, a high antigen density enhances the internalization and intracellular delivery of the cytotoxic payload. Consistency in target antigen expression across different patients with the same type of cancer is also an important consideration for the broader application of the ADC [22]. Antigens that are secreted on the outside of the target cell may reduce the binding rate between the ADC and the target cell and potentially posing safety concerns. Thus, a target antigen for ADC should ideally be located on the cell’s surface or extracellular region while ADC circulates within the body [15,23]. Furthermore, the payload must facilitate efficient cellular internalization, as well as effective cleavage and release upon ADC binding to the target antigen [24]. Conventionally, full-length IgG1 monoclonal Antibodies, which are abundant in serum, are used for ADCs development. These IgG1 Antibodies possess valuable properties such as high binding affinity to Fc receptor-binding regions, antibody-dependent cell-mediated cytotoxicity (ADCC), phagocytosis, and complement cytotoxicity [25].

Non-specific uptake of antibodies can give rise to side effects and reduce specificity. Striking a balance between achieving the desired therapeutic effects and potential non-specific interactions is a critical factor in the development and application of ADC-based therapies. On the one hand, Fc receptors, typically engaged by antibodies, are expressed on various immune cells, notably macrophages and dendritic cells. However, interactions between Fc receptors and antibodies can lead to non-specific binding and internalization, potentially triggering inflammatory responses and contributing to the side effects and adverse reactions associated with antibody-based therapies. On the other hand, lectin receptors on various immune cells, including macrophages and dendritic cells, have the capacity to recognize and bind to carbohydrate structures present on glycoproteins and glycolipids, potentially resulting in non-specific antibody uptake. Hence, strategies aimed at minimizing non-specific uptake while enhancing specificity are of paramount importance in the development and utilization of ADCs [26].

However, their relatively large molecular weight (~150 kDa) can impede transportation and binding to solid cancers ensconced in the cancer microenvironment, characterized by poor blood supply and a pericellular matrix [27,28]. To surmount these challenges, an innovative ADC has been devised, utilizing solely the fragment antigen-binding region (Fab) of the antibody while excluding the Fc region. This reduction in molecular weight enables ADC to navigate through barriers in the cancer microenvironment, thereby enhancing its effectiveness in treating solid cancers [29]. Additionally, ongoing research explores the development of novel Antibodies targeting a variety of antigens through the utilization of human single-domain (ScFv) antibody libraries [30].

### 2.2. Linker

The linker serves as the crucial bridge between the antibody and the payload. Its primary role involves maintaining binding while circulating in the bloodstream and subsequently releasing the payload upon internalization into the target cell. This pivotal role contributes significantly to the overall stability of the ADCs [31]. specific nature of the linker holds sway over the pharmacokinetics and pharmacodynamics of the ADCs, with the site of conjugation between the linker and the antibody also wielding influence over payload effectiveness [32,33]. Achieving the right balance between stability and release capability hinges on factors like linker length, type, and conjugation site. Consequently, for optimal ADCs efficacy, the design of linkers must prioritize both robust stability and efficient payload release mechanisms [32].

Following internalization into target cells, ADCs undergoes a sequence of cleavage events, starting with the breakdown of the antibody and linker under specific conditions [34] (Figure 3). Subsequently, the linker and payload are cleaved by lysosomes. The hydrophobic nature of the cleaved payload enables it to traverse the target cell’s membrane, giving rise to a bystander effect that not only eliminates the target cell but also neighboring cells. On occasion, linker cleavage may occur in the bloodstream, leading to a similar bystander effect [13]. Two distinct categories of cleavable linkers exist: chemically cleavable linkers, sensitive to factors like acid or glutathione concentrations, and enzyme cleavable linkers. Acid-sensitive cleavable linkers are activated within the acidic environments of endosomes (pH 5–6) and lysosomes (pH 4.8) upon internalization [12]. For instance, Mylotarg, used in relapsed acute myeloid leukemia, features a CD33 antibody comprising an acid-sensitive cleavable linker and a DNA-damaging agent (N-acetyl-γ-calicheamicin) [35,36]. However, they come with the limitation of potentially releasing highly cytotoxic payloads even when internalized by normal cells instead of target cells [13,14]. Glutathione-sensitive cleavable linkers react when glutathione levels are elevated. This leverages the fact that cells secrete glutathione during stress, and cancer cells produce more glutathione than normal cells [11]. An example is lorvotuzumab mertansine, a multiple myeloma drug currently undergoing clinical trials. It incorporates a CD56 antibody with a glutathione-sensitive cleavable linker and a tubulin inhibitor (Maytansinoid DM1) [37,38]. Additional enzyme cleavable linkers encompass peptide linkers, sulfatase linkers, and phosphatase linkers. Peptide linkers rely on proteases for cleavage, with the FDA-approved Hodgkin’s lymphoma drug brentuximab vedotin using an CD30 antibody with a peptide cleavable linker and MMAE [39]. Furthermore, various enzyme-based linkers, such as β-glucuronidase and β-galactosidase linkers, have been developed [40,41,42,43] (Figure 4). These are degraded by enzymes present in lysosomes, rendering them more stable in the bloodstream compared to chemically cleavable linkers.

Non-cleavable linkers, which remain intact within the lysosome post-internalization and do not undergo degradation, lead to antibody degradation by lysosomal peptidases. This, in turn, results in the release of payload and triggers cytotoxicity, causing cell death (Figure 3). Unlike their cleavable counterparts, non-cleavable linkers do not induce a bystander effect on neighboring cells in the bloodstream where peptidases are absent. Additionally, they offer enhanced plasma safety and greater resistance to proteolysis compared to cleavable linkers. Recent advancements have given rise to non-cleavable linkers, notably the SMCC (Succinimidyl-4-(N-maleimidomethyl)cyclohexane-1-carboxylate) linkers and MD (Molecular Dynamics) linkers, which are currently in development. The hydrophilic nature of linker plays a pivotal role in diminishing nonspecific uptake or off-target toxicity of payload. SMCC linkers incorporate a cyclo-nucleic acid ring that reduces the antibody conjugate’s hydrolysis rate at high pH levels, thus bolstering stability while preserving antibody specificity. On the other hand, MD linkers substitute the cyclo-nucleic acid ring of SMCC linkers with 1,3-dioxolein, enhancing linker hydrophilicity and stability due to the presence of two oxygen atoms within the 1,3-dioxolein ring (Figure 4). Notably, the current ADCs technologies making use of this approach include Kadcycla^®^ (Basel, Switzerland) [44,45,46].

### 2.3. Payload

The primary role of payload entails internalizing ADC within the target cell and subsequently releasing it from ADC to trigger cell death in the target. Early ADCs employed drugs like doxorubicin and vinblastine, which possessed relatively low cytotoxicity, resulting in limited efficacy against cancer cells. The utilization of more potent cytotoxic agents within ADCs led to heightened toxicity and a bystander effect impacting normal cells as well. Therefore, payload needs to effectively eliminate most target cells, even at low concentrations in the nanomolar or picomolar range. It should also restrict the number of payloads binding to antibodies to prevent antibody inhibition and maintain stability during circulation within the body. Instable payloads are vulnerable to degradation during conjugation or storage [13,47]. Additionally, payloads should exhibit low immunogenicity since they play a crucial role in the tumor immune microenvironment and can influence the immune response. In the context of the multiple myeloma drug Belantamab Mafodotin, the payload Monomethyl auristatin F (MMAF) promotes dendritic cell activation in vivo [48,49].

Current ADCs payloads encompass two main categories: DNA-damaging agents and microtubule polymerization inhibitors, both of which play critical roles in disrupting cancer cells. DNA-damaging agents such as calicheamicin, duocarmycin, and pyrrolobenzodiazepine (PBD) interact with DNA, leading to cleavage and alkylation. These agents bind to the minor groove within DNA via directional hydrogen bonding and van der Waals interactions, causing structural disruption and cell death. For instance, calicheamicin induces a reaction akin to bergman cyclization, generating a diradical species that cleaves the DNA strand, while duocarmycin triggers irreversible DNA alkylation, disrupting DNA integrity and causing cell demise. DNA SJG-136 binds to DNA’s guanine sequence, forming bridges between existing DNA bonds, further disrupting the DNA structure and leading to cell death [50,51]. PBDs insert into DNA’s minor grooves and perform nucleophilic attacks, interfering with DNA processing and culminating in cell death. Microtubule polymerization inhibitors have five binding sites and fall into two categories: those stabilizing or destabilizing microtubule structure. For example, paclitaxel promotes tubulin polymerization and stabilizes its structure, preventing spindle formation in cell division, thereby arresting cancer cells in G2 and M phases. Conversely, agents like mertansine and auristatin inhibit tubulin polymerization and destabilize its structure by forming cross-links between tubulin [52]. Among the auristatin-based payloads with anticancer activity are monomethyl auristatin E (MMAE) and monomethyl auristatin F (MMAF). MMAE, a peptide analog with five amino acids, is significantly more potent than vinca alkaloids, but MMAF, with a negatively charged C-terminal carboxylic acid group, exhibits advantages in ADCs manufacturing and drug delivery due to its hydrophilicity. MMAF’s negatively charged group restricts its diffusion in cells, making it less toxic than MMAE. Both MMAF and MMAE penetrate cell membranes and induce a bystander effect, diffusing to neighboring cells. When an ADC, consisting of an antibody-linker (valine, citrulline), enters a cell, cathepsin cleaves the payload, inhibiting microtubule binding and polymerization, ultimately leading to target cell death [50,53,54]. Maytansinoid DM1, derived from the African shrub maytenus ovatus in 1972, binds to microtubules, causing cell cycle arrest and apoptosis. DM1 strongly inhibits dynamic instability parameters like microtubule growth and shortening rates at sub-nanomolar concentrations, making it highly effective in killing cancer cells. However, DM1 lacks cancer specificity and exhibits systemic toxicity, limiting its use as a standalone anticancer agent [51,55,56]. The breast cancer drug trastuzumab emtansine comprises HER2 antibody-SMCC-DM1 [21].

Innovative non-cytotoxic payloads in ADCs, known as immune-stimulating antibody conjugates (ISACs), merge the precision of antibody-guided targeting with the immunomodulatory capabilities of small molecules. On one hand, antibodies conjugate immuno-agonist small molecules to bolster the immune response, while tumor-specific targeting reduces the potential toxicity of these molecules. Currently, there is a range of immune-modulating ADC drugs in development, with emerging payloads including Toll-like receptor (TLR) agonists, stimulators of the interferon gene (STING), and glucocorticoid receptor modulator. Within the TLR family, TLR7, TLR8, and TLR9 agonists are getting attention for their ability to activate dendritic cells and enhance T cell-mediated tumor cell killing. Combining STING agonists with immune checkpoint inhibitors holds great promise for significantly augmenting the effectiveness of immunotherapy and represents an innovative treatment approach. Glucocorticoids can act as ADC payloads to treat autoimmune and inflammatory diseases [57,58,59].

Besides these currently available payloads, ongoing research is developing payloads targeting chaperones, splicing, proteasomes, and more [60,61,62,63,64]. ADCs, designed to enhance diversity in target cells, reduce side effects, and boost cytotoxicity by linking to two payload antibodies, are also under development [59,65,66].

### 2.4. Conjugation

The process of connecting linker-payloads, either by utilizing naturally occurring amino acids on the antibody surface or through artificial modifications, is referred to as conjugation. Conjugation can be categorized into stochastic conjugation and site-specific conjugation based on the control over the conjugation site. Given that the conjugation site significantly influences the binding capacity, internalization, and stability of the monoclonal antibody and its antigen, careful consideration is required when selecting a conjugation site to avoid sensitive areas such as antigen recognition sites and FC receptor binding sites.

Following conjugation, ADC are assigned a Drug Antibody Ratio (DAR) unit, which is a numerical value ranging from 0 to 8, indicating the number of drugs attached to a single monoclonal antibody. Higher DARs exhibit increased cytotoxicity against target cells. However, ADC with higher DARs possess larger sizes and harbor hydrophobic payloads, making them more prone to aggregation and faster clearance from the bloodstream [67,68]. Consequently, ADCs with DARs ranging from 2 to 4 are commonly used. Additionally, ADCs with the same DAR can be categorized into two groups: homogeneity ADCs and heterogeneity ADCs, based on variations in the junction sites. Heterogeneity ADCs, due to their lack of control over junction sites, exhibit lower efficacy and more pronounced side effects compared to homogeneity ADCs [69].

Stochastic conjugation is a technique that uses naturally occurring lysine and cysteine amino acids found in monoclonal antibodies as conjugation sites, followed by a chemical reaction to create an ADC. These amino acids selected as conjugation sites are abundant, facilitating convenient conjugation. However, a drawback of this method is the inability to avoid random conjugation, resulting in the generation of heterogeneous ADCSs. To address this, NHS ester is used to modify over 40 of the approximately 85 lysines present in IgG1 antibodies. This modification enhances reactivity and produces stable ADCs [70,71,72]. In the case of Pfizer (New York, NY, USA) /Wyeth’s (Madison, NJ, USA) MyLotarg^®^, which utilizes lysine as a conjugation site, it achieves a DAR of 3 [73]. On the other hand, cysteine is utilized for conjugation through disulfide bonds, with 12 binding sites located internally within the monoclonal antibody and 4 on its surface. An electrophilic compound, maleimide, is employed to break the disulfide bonds, resulting in the creation of heterogeneous ADCs with DAR values ranging from 0 to 8 [74]. Cysteine is known for being a non-specific and unstable conjugation site due to its highly reactive thiol (-SH) group and its involvement in antibody binding. Consequently, methods have been developed to substitute cysteine with serine to produce ADCSs with DAR values of 2 or 4 [75,76].

Site-specific conjugation is distinguished by its precision in performing chemical reactions, utilizing enzymes, and implementing glycan modifications exclusively at the designated conjugation site, resulting in the creation of a homogeneous ADCs. One approach to achieving site-specific conjugation involves incorporating cysteine at a specific location within a monoclonal antibody. The THIOMAB technique, for instance, involves strategically placing cysteine on the heavy and light chains of an antibody, allowing for the production of homogeneous ADCSs with a DAR of 2 while preserving intra-chain bonds [77]. Another method using cysteine for site-specific conjugation is disulfide rebridging, where drugs are bound after reducing intra-chain bonds. When bisulfone is used for rebridging, disulfide reconnection occurs through three carbon bridges, generating an ADC with a DAR of 4 [78]. Site-specific conjugation also encompasses various methods, including those utilizing N/C termini, unnatural amino acids (UAAs), transglutaminase, and glycan modification. When targeting the N/C termini, conjugation involves attaching the drug to the N or C terminus found within the monoclonal antibody to create homogeneous ADCs. Care should be exercised, especially with the N terminus, as it is in close proximity to the antibody’s antigen-binding site [79,80]. Unnatural amino acids (UAAs) can be inserted into the antibody sequence to facilitate drug conjugation, leveraging the unique functional groups of these UAAs. However, since UAAs can potentially induce immunogenicity, it is advisable to use structural analogs of standard amino acids, such as p-acetylphenylalanine [81,82,83]. Bacterial-derived transglutaminases offer substrate-specificity and catalytic effects, enabling the binding of payloads to antibodies in a precise manner [84,85]. Glycan-modified conjugation takes advantage of the glycoprotein nature of antibodies, with the glycosylation site present on the antibody’s HC-N297 being conserved in most cases and situated away from the antigen recognition site. Drug conjugation to the glycosylation site can be achieved through glycan oxidation or by using enzymes like endoglycosidase and β-1,4-galactosyltransferase [86,87]. The development of AJICAP, a site-specific conjugation technology, has led to the production of low heterogeneity ADCs. Precise analysis using Strong Cation Exchange Chromatography coupled with UV and mass spectrometry (SCX-UV-MS) allows for the characterization of drug-antibody ratios and charge variants in a single assessment, confirming the Fc site selectivity achieved with AJICAP conjugation. This technology facilitates the creation of consistent batches and the advancement of new analytical methods [88]. In recent advancements in Site-Specific Conjugation, AJICAP was used to modify native antibodies at Lys248, leading to the development of ADCs with a significantly expanded therapeutic index compared to the FDA-approved Kadcyla. This innovative approach eliminates the need for redox treatment, bolsters the stability of Fc affinity reagents, and enables the production of diverse ADCs without encountering issues related to aggregation. Furthermore, AJICAP’s versatility was demonstrated by facilitating the creation of various ADCs, including antibody-protein and antibody-oligonucleotide conjugates, underscoring its potential for site-specific antibody conjugate manufacturing without the necessity of antibody engineering [89].

## 3. US FDA Approved ADC and Market Status

The global market share of ADC R&D and therapeutics is growing rapidly, with eight ADCs receiving FDA approval for therapeutic use from 2019 to 2022 [10,90]. The human epidermal growth factor receptor 2 (HER2) and trophoblast cell-surface antigen 2 (TROP2) targets acco [91,92]. Of the 13 ADCSs approved by the FDA, 12 are commercially available after Blenrep^®^ (London, UK)was withdrawn due to safety concerns (Table 1). In 2022, overall ADC sales grew 32% from the previous year to $7.927 billion, with the ovarian cancer drug ELAHERE^®^ (Waltham, MA, USA), approved in November 2022, generating sales of $2.55 million and sales of $76.9 million during the second quarter of 2023. Tivdak^®^ (Copenhagen, Denmark) for cervical cancer generated $17.82 million in sales in 2022, a 191% increase from last year. Zynlonta^®^ (Epalinges, Switzerland) for large B-cell lymphoma generated $19.62 million in sales in 2022, up 16.5% year-on-year. Blenrep^®^ (London, UK) for myeloma generated $147.4 million in sales in 2022, up 33% year-on-year. Trodelvy^®^ (Foster, CA, USA) for breast cancer generated $673.9 million in sales in 2022, up 78.9% year-on-year. Enhertu^®^ (Cambridge, UK) for breast cancer saw the largest revenue increase, with $1290.3 million in sales in 2022, up 189.7% year-on-year, driven by expanded treatment indications. Padcsev^®^ (Tokyo, Japan) for urothelial carcinoma generated $121.5 million in sales in 2022, up 32% year-on-year. Polivy^®^ (Basel, Switzerland) for large B-cell lymphoma generated $587.2 million in sales in 2022, up 81.3% year-over-year. Lumoxiti^®^ (Cambridge, UK), a treatment for hairy cell leukemia, posted a loss of $104,830 in 2022. Besponsa^®^ (New York, NY, USA) for lymphoblastic leukemia generated $217.6 million in sales in 2022, up 14% year-on-year. Kadcsyla^®^ (Basel, Switzerland) for breast cancer generated $2.254 billion in sales in 2022, up 7% year-on-year. Adcetris^®^ (Tokyo, Japan) for lymphoma generated $236.47 million in sales, up 35% year-on-year. Mylotarg^®^ (New York, NY, USA) for acute myeloid leukemia did not show significant sales compared to other ADCs products. The pipeline of new clinical ADCs in 2022 is 211, with 75% in solid tumor and 25% in hematologic cancer indications. Targets are dominated by HER2 and TROP2, and payloads are diverse, with microtubule inhibitors predominantly used but decreasingly so [93,94] (Figure 5).

### 3.1. Mirvetuximab Soravtansine (ELAHERE^®^—ImmunoGen)

Mirvetuximab soravtansine was approved by the FDA in 2022 for the ADCs treatment of peritoneal cancer, platinum-resistant epithelial ovarian cancer, and fallopian tube cancer. Mirvetuximab soravtansine targets the folate receptor alpha (FRα) and its payload utilizes DM4 and a cleavable disulfide linker, Sulfo-SPDB. It binds to FRα on the surface of epithelial tumor cells in ovarian or fallopian tube cancer, and after the internalization process occurs, DM4 is released, inducing cell death by cell cycle arrest and spreading to surrounding cells to cause more cell death [95,96].

### 3.2. Tisotumab Vedotin-Tftv (Tivdak^®^—Seagen)

Tisotumab vedotin is a cervical cancer treatment approved by the FDA in 2021. It targets tissue factor (TF-011) and the payload utilizes MMAE and a cleavable valine-citrulline, MC-Val-Cit-PABC linker. MMAE is delivered intracellularly to block tubulin polymerization and halt cell division [97,98].

### 3.3. Loncastuximab Tesirine-Lpyl (Zynlonta^®^—ADC Therapeutics)

Loncastuximab tesirine is a treatment for diffuse large B-cell lymphoma that was approved by the FDA in 2021. It targets CD19 and the payload utilizes a PBD Dimer and a Val-Ala dipeptide linker, which is a cleavable enzymatic type linker. It exerts a potent cytotoxic effect by promoting the formation of cross-links in DNA and subsequent arrest of cell division [64,99].

### 3.4. Belantamab Mafodotin-Blmf (Blenrep^®^—GlaxoSmithKline)

Belantamab mafodotin-blmf is a treatment for multiple myeloma that was approved by the FDA in 2020. It targets B-cell maturation antigen (BCMA) and the payload is MMAF with a non-cleavable maleimidocaproyl (MC) linker. The ADC complex binds to BCMA, internalizes into the cell, and triggers cell cycle arrest and apoptosis via MMAF [100,101].

### 3.5. Sacituzumab Govitecan (Trodelvy^®^—Immunomedics)

Sacituzumab govitecan is a breast cancer drug approved by the FDA in 2021. It targets Trop-2 and the payload utilizes SN38 and a carbonate linker, which is a hydrazone cleavable linker. It induces DNA damage and consequently cell cycle arrest. Due to its membrane-permeable ability, it can induce an apoptotic effect in neighboring cells [102,103].

### 3.6. Trastuzumab Deruxtecan (Enhertu^®^—Daiichi Sankyo/AstraZeneca)

Trastuzumab deruxtecan is a treatment for metastatic breast cancer that was approved by the FDA in 2022. It targets HER2 and uses a camptothecin payload that functions as a TOP1 inhibitor and a cleavable tetrapeptide-based linker. It binds to HER2 on the surface of cancer cells and undergoes internalization, where the linker is cleaved by cathepsin B, releasing the payload into the cytoplasm and killing tumor cells. Unlike trastuzumab emtansine, it uses a payload that can easily pass through cell membranes to deliver cytotoxic effects to target cells and neighboring cancer cells [20,104].

### 3.7. Enfortumab Vedotin (Padcev^®^—Astellas/Seagen Genetics)

Enfortumab vedotin is a treatment for urothelial carcinoma that was approved by the FDA in 2021. It targets the cell adhesion molecule Nectin-4, which is overexpressed in 97% of urothelial carcinomas. The payload uses MMAE and a cleavable valine-citrulline, MC-Val-Cit-PABC linker. Intracellularly, it causes cell cycle arrest and apoptosis [105,106].

### 3.8. Polatuzumab Vedotin-Piiq (Polivy^®^—Genentech/Roche)

Polatuzumab vedotin is a treatment for large B-cell lymphoma that was approved by the FDA in 2019. It targets CD79b and the payload uses an MC-Val-Cit-PABC linker, which is an MMAE and cleavable valine-citrulline. Intracellularly, it causes cell cycle arrest and apoptosis [107].

### 3.9. Inotuzumab Ozogamicin (Besponsa^®^—Pfizer/Wyeth)

Inotuzumab ozogamicin is a treatment for B-cell acute lymphocytic leukemia that was approved by the FDA in 2017. It targets CD22 and its payload uses N-acetyl-γ calicheamicin and an acid-labile hydrazone linker. After internalization into the cell, calicheamicin binds to double-stranded DNA and causes cell death [108,109].

### 3.10. Trastuzumab Emtansine (Kadcyla^®^—Roche)

Trastuzumab emtansine (T-DM1) is an ADC treatment for metastatic breast cancer that was approved by the FDA in 2013 and is the top-selling ADC among all ADCs with sales of approximately $2.265 billion in 2022. It targets HER2 and uses DM1 as a payload and non-reducible thioether as a linker. After binding to HER2 on the surface of cancer cells and entering the cell through internalization, active DM1 is released. The released DM1 inhibits microtubule assembly, leading to cellular mitotic arrest and apoptosis [110,111].

### 3.11. Brentuximab Vedotin (Adcetris^®^—Seagen/Takeda)

Brentuximab vedotin was approved by the FDA in 2011 for the treatment of systemic anaplastic large cell lymphoma (ALCL), Hodgkin lymphoma, and cutaneous T-cell lymphoma. It targets CD30 and the payload uses MMAE and a cleavable valine-citrulline, MC-Val-Cit-PABC linker. It binds to CD30 on the tumor epithelium, undergoes internalization, and cleaves the linker, releasing MMAE, which kills tumor cells by interfering with microtubule polymerization [112,113].

### 3.12. Gemtuzumab Ozogamicin (Mylotarg^®^—Pfizer/Wyeth)

Gemtuzumab ozogamicin is a treatment for acute myeloid leukemia that was approved by the FDA in 2017. It targets CD33 and the payload utilizes N-acetyl-γ calicheamicin with an acid-labile hydrazone linker. Calicheamicin binds to double-stranded DNA and causes cell death [19].

## 4. Research Trends in ADCs

After analyzing 71 research papers published within the past five years on ADCS, we have identified a predominant focus on 36 different target antigens and 15 distinct payloads (Figure 6 and Appendix A). The most extensively studied target antigen, appearing in 40% of the papers, is human epidermal growth factor receptor type 2 (HER2). HER2 is predominantly expressed in breast cancer and is also frequently found in other cancer types, including ovarian, uterine, gastric, and lung cancers [114,115]. A notable monoclonal antibody targeting HER2 is Trastuzumab, which is utilized in ADCs like Trastuzumab Deruxtecan and Trastuzumab Emtansine [116,117]. The next highly researched target antigen is the Epidermal Growth Factor Receptor (EGFR), which is primarily overexpressed in solid tumors such as lung, pancreatic, and colorectal cancers [118]. ADCs targeting EGFR include M1231 and Depatuxizumab mafodotin [119,120]. Another extensively studied target is the B7 homolog 3 protein (B7-H3) transmembrane protein, which exhibits rare expression in normal tissues but is frequently overexpressed in solid tumors. A representative ADCs associated with B7-H3 is Ifinatamab deruxtecan [121,122]. Tumor-associated calcium signal transducer 2 (TROP-2), a cell surface receptor involved in intracellular calcium signaling, is the fourth well-researched target. It is highly expressed in pancreatic, cervical, gastric, and cholangiocarcinoma. Representative ADCs for TROP-2 include Sacituzumab govitecan and Datopotamab deruxtecan [123,124]. Additionally, there is ongoing development of target antigens like c-Met (mesenchymal-epithelial transition factor), a hepatocyte growth factor receptor, human epidermal growth factor receptor type 3 (HER3), chondroitin sulfate proteoglycan 4 (CSPG4), which plays a role in stabilizing early melanoma cells, and CD30, a tumor necrosis factor receptor and tumor marker, for various ADCS applications [125,126,127].

Regarding linkers, the current research and development landscape primarily focuses on 64 cleavable linkers and 11 non-cleavable linkers. Cleavable linkers encompass various types, including acid cleavable, GSH cleavable, Fe(II) cleavable, cathepsin cleavable, glycosidase cleavable, phosphatase cleavable, and sulfatase cleavable. Non-cleavable linkers include SMCC linker and MD linker. Recently, photo-responsive cleavable and bioorthogonal cleavable linkers have emerged alongside non-cleavable Mal-PAB linkers.

Among the studied payloads, auristatin-based and mertansine-based payloads dominate the field. Auristatin-based payloads, including MMAE and MMAF, are extensively employed. Auristatin, derived from Dolastatin 10 isolated in Dolabella auricularia, exhibits potent proliferation inhibition across various cancer types. It induces apoptosis by disrupting tubulin assembly and inducing cell cycle arrest. Mertansine-based payloads like DM1 and DM4 possess remarkable cytostatic activity at specific concentrations. However, they have associated toxic side effects on the gastrointestinal tract and nerve cells due to a lack of target specificity. Research efforts are underway to enhance target specificity through ADC conjugation to mitigate these side effects [118,120]. Furthermore, ADC research and development are exploring payloads such as topoisomerase 1 (TOP 1), calicheamicin, and tubulysin. TOP 1 is an enzyme involved in DNA replication and transcription and has been extensively studied as a target for solid tumors, including studies involving ABBV-400 [128]. Calicheamicin, recently approved in the U.S., is primarily used in leukemia and small cell lung cancer treatment. Upon contact with DNA, calicheamicin undergoes robust Bergman cyclization and cleaves DNA to eliminate tumor cells [116]. Tubulysin, a tetrapeptide isolated from myxomycetes culture medium, similar to the auristatin-based payload MMAE, binds to the vinca alkaloid binding site on tubulin, inhibiting tubulin polymerization and leading to cancer cell death. Tubulysins exhibit potent cytotoxicity against various cancer types, including breast, colon, lung, ovarian, and prostate cancers, making them the focus of extensive research and development [129].

## 5. Current Clinical Progress of ADCs

After conducting searches on clinicaltrial.gov, EU clinical trials register, Health Canada clinical database, and Google Scholar, we have compiled a summary of 128 clinical studies involving ADCs from 2018 to 2023 (Figure 7 and Appendix A). The predominant target antigen under investigation is HER2, with a significant number in phase II trials, and 97 in phase I-II trials. These ADCs employ a combination of 111 cleavable linkers, 4 non-cleavable linkers, and 13 unknown linkers. The most frequently used linker is Valine-Citrulline (Val-Cit). In terms of payloads, there are 90 studies focusing on microtubule polymerization inhibitors, 31 exploring DNA damage agents, with the most commonly used payload being MMAE (Figure 8).

## 6. Conclusions

In conclusion, the field of anticancer therapeutics has witnessed significant advancements in recent years, with a particular focus on the development of Antibody-Drug Conjugates (ADCs) as a promising approach. These ADCs consist of a three-component structure: an antibody that selectively binds to a target antigen, a highly potent drug (payload) that induces targeted cell death, and a linker that connects the antibody to the drug. This combination offers several advantages, including high target specificity, prolonged half-life in the body, and reduced side effects compared to traditional chemotherapy.

The evolution of ADCs can be categorized into three generations, each addressing specific limitations and challenges. First-generation ADCs faced issues related to drug binding sites on antibodies, efficiency, and side effects. Second-generation ADCs employed advanced techniques, such as controlling the drug-antibody ratio (DAR) and selective conjugation of drugs to specific antibody sites, to enhance their efficacy and safety. The latest third-generation ADCs aim for uniform drug binding to antibodies without compromising antigen-antibody binding.

The growth of ADCs aligns with the rising prominence of biopharmaceuticals in the pharmaceutical market, with anticancer drugs projected to dominate global drug consumption. In particular, the ADC market is expected to expand significantly, reaching multi-billion-dollar figures. The recent surge in ADCs candidates and approvals reflects the growing interest and investment in this therapeutic approach.

Research efforts have predominantly focused on targeting antigens like HER2 and EGFR, showing promise in various cancers. Clinical studies have further demonstrated the potential of ADCs, with a substantial number of trials in progress, particularly for HER2-targeted ADCs. The choice of linkers and payloads in ADCs design plays a critical role in their stability and efficacy, with Val-Cit linkers and microtubule polymerization inhibitors being common choices.

In summary, Antibody-Drug Conjugates represent a rapidly evolving and promising avenue in the field of anticancer therapeutics. As research continues to uncover novel targets, linkers, and drug payloads, the potential for ADCs to revolutionize cancer treatment and improve patient outcomes remains high. The future of anticancer therapeutics is undeniably intertwined with the continued growth and innovation of ADCs.

## Figures and Tables

**Figure 1 antibodies-12-00072-f001:**
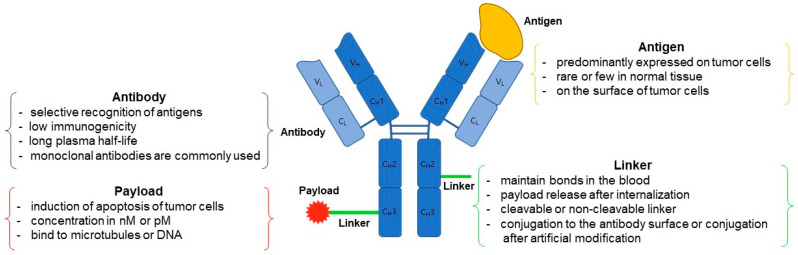
The structure and components of an ADC. ADC consist of a monoclonal antibody, a linker, and a cytotoxic payload. Antibody comprises two heavy chains and two light chains, containing four disulfide bonds within the heavy chains and one each in both the heavy and light chains. The linker-payload is connected to the external surface of the antibody, ensuring both easy attachment to the antibody and effortless detachment from the target cell.

**Figure 2 antibodies-12-00072-f002:**
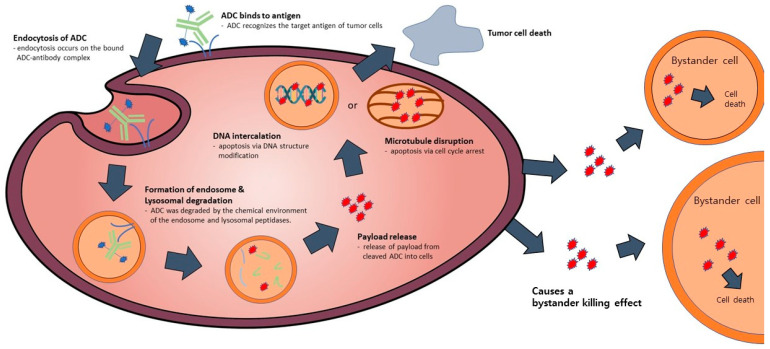
The mode of action of ADC. The ADC attaches to the receptor located on the target cell, resulting in the formation of an ADC-antigen complex. This complex subsequently undergoes internalization by endocytosis, forming endosomes, and goes through maturation where it is degraded by lysosomes. Throughout this process, the activated payload binds to target DNA or microtubules, leading to cell death induction. Additionally, intracellular drugs have the potential to exit the cell and induce a bystander effect, resulting in cell death in neighboring cells.

**Figure 3 antibodies-12-00072-f003:**
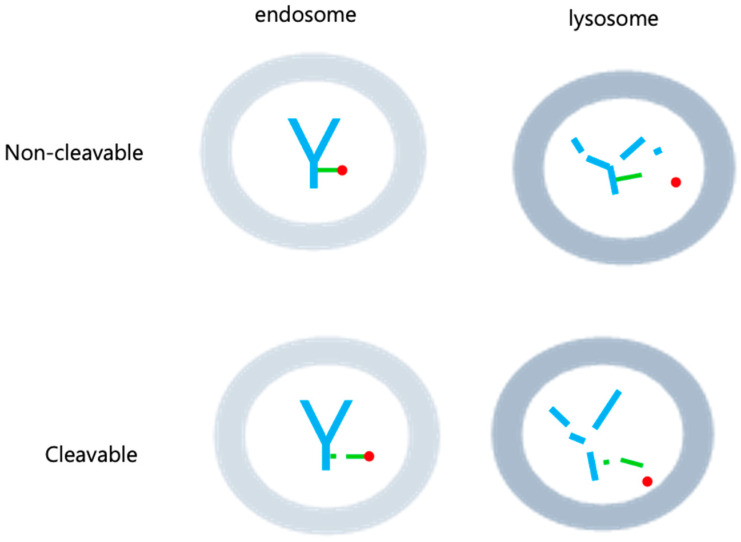
Cleavable and Non-cleavable linkers. Cleavable linkers have the ability to break down in specific situations, like acidic environments, varying glutathione levels, or the influence of enzymes. This breakdown can happen within both endosomes and lysosomes. On the other hand, non-cleavable linkers are not susceptible to degradation in either endosomes or lysosomes, as the linker itself remains intact. Only the payload is liberated within the lysosome. In the figure, antibody are shown in blue, linker in green, and drug in red.

**Figure 4 antibodies-12-00072-f004:**
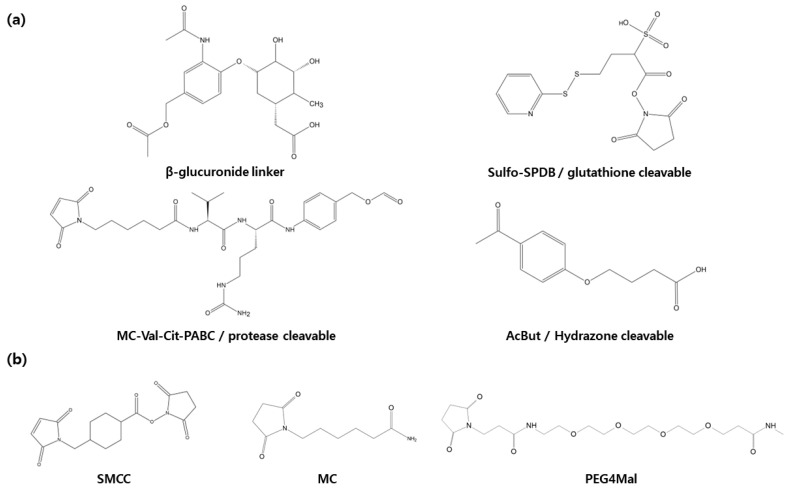
In the development of antibody-drug conjugates linkers play a crucial role. (**a**) Cleavable linkers include β-glucuronide linker, glutathione linkers, protease linkers, hydrazone linkers. (**b**) Non-cleavable linkers include SMCC, MC, PEG4Mal.

**Figure 5 antibodies-12-00072-f005:**
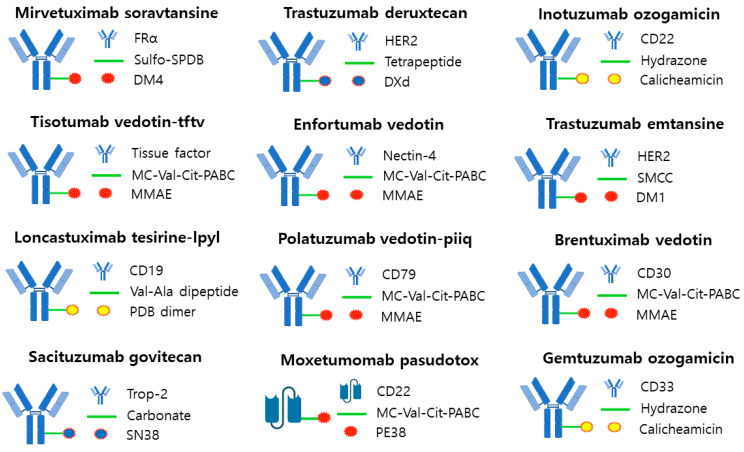
FDA-approved ADC. The majority of linkers utilized in ADCs are cleavable, although trastuzumab etansine used a noncleavable linker. The payloads used consisted of three types: a DNA-targeting agent (yellow), a tubulin-binding agent (red), and a topoisomerase 1 inhibitor (blue).

**Figure 6 antibodies-12-00072-f006:**
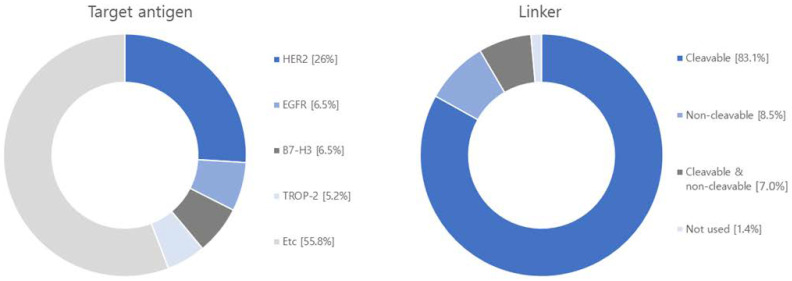
Diagram illustrating target antigens, linkers, payloads and diseases used in research articles. Among the 71 articles, the target antigen is HER2 in 20, EGFR in 5, B7-H3 in 5, TROP-2 in 4. As for the linker, 64 are cleavable, and 11 are non-cleavable, and 1 not-used linker. Regarding the payload, 30 articles focus on MMAE, 10 on DM1, and 9 on Topoisomerase 1 inhibitor. Additionally, there are 6 articles discussing DXd and 6 on PBD. The research covers various diseases in the following order: breast cancer with 29 articles, lung cancer with 18 articles, ovarian cancer with 12 articles, pancreatic cancer with 10 articles, colorectal cancer with 10 articles, stomach cancer with 6 articles, and melanoma with 6 articles.

**Figure 7 antibodies-12-00072-f007:**
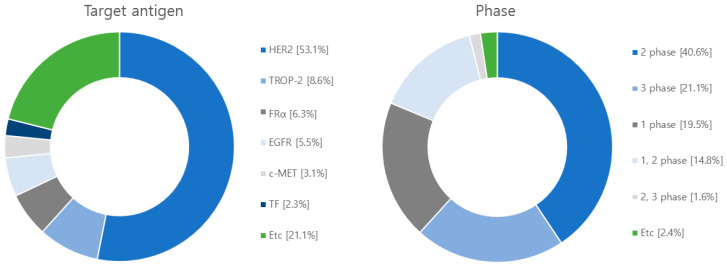
Diagram illustrating target antigens and clinical phases employed in clinical trials. Out of the 128 clinical investigations conducted, HER2 was the targeted antigen in 68 studies, while TROP-2 was the focus in 11 studies. Additionally, FRα was targeted in 8 studies, EGFR was targeted in 7 studies followed by c-MET in 4 studies. The remaining 30 studies targeted other antigens. In terms of clinical stages, 52 studies were categorized in phase 2, 25 in phase 1, 19 in both phases 1 and 2, 27 in phase 3, and 2 in the combined phases 2 and 3.

**Figure 8 antibodies-12-00072-f008:**
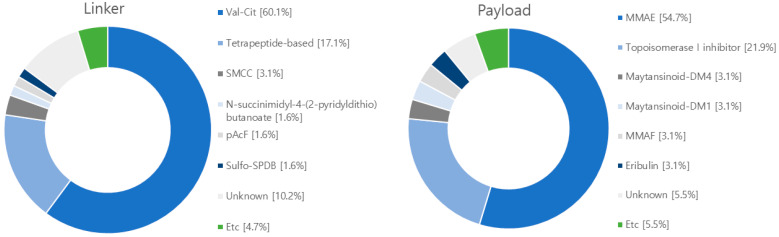
Diagram illustrating linkers and payloads used in clinical trials. The linker’s composition includes 77 instances of Val-Cit cleavable linker, 22 instances of tetrapeptide-based linker, 2 instances of SPDP linker, 1 instance of sulfo-SPDB linker, and 1 instance of glucuronide-trigger linker. Meanwhile, the non-cleavable linkers comprise 4 instances of SMCC linker and 13 instances of linkers with unknown attributes. Within the payload category, microtubule inhibitors consist of 70 cases of MMAE, 4 cases of eribulin, 4 cases of maytansinoid-DM4, 4 cases of MMAF, 4 case of maytansinoid-DM1, and 1 case of hemiasterlin. In relation to DNA damage reagents, there are 28 instances of topoisomerase I inhibitor, 1 instance of duocarmacin, and 7 instances of payloads with unknown characteristics.

**Table 1 antibodies-12-00072-t001:** List of commercially available ADCs with FDA approval.

ADC	Registered Trademark	Company	Disease	Antigen	Linker	Payload	Approved Year
Mirvetuximab soravtansine	ELAHERE	ImmunoGen (Waltham, MA, USA)	Platinum-resistant epithelial ovarian	FRα	Sulfo-SPDB	DM4	2022
Tisotumab vedotin-tftv	Tivdak	Seagen Inc (Copenhagen, Denmark)	Recurrent or metastatic cervical cancer	Tissue factor	MC-Val-Cit-PABC	MMAE	2021
Loncastuximab tesirine-lpyl	Zynlonta	ADC Therapeutics (Epalinges, Switzerland)	Diffuse large B-cell lymphoma	CD19	Val-Ala dipeptide	PDB dimer	2021
Belantamab mafodotin-blmf	Blenrep	GlaxoSmithKline (London, UK)	Relapsed or refractory multiple myeloma	BCMA	MC	MMAF	2020; withdrawn 2022
Sacituzumab govitecan	Trodelvy	Immunomedics (Foster, CA, USA)	Metastatic triple-negative breast cancer	Trop-2	Carbonate	SN38	2020
Trastuzumab deruxtecan	Enhertu	AstraZeneca/Daiichi Sankyo (Cambridge, UK)	Unresectable or metastatic HER2-positive breast cancer	HER2	Tetrapeptide	DXd	2019
Enfortumab vedotin	Padcev	Astellas/Seagen Genetics (Tokyo, Japan)	Advanced or metastatic urothelial carcinoma	Nectin-4	MC-Val-Cit-PABC	MMAE	2019
Polatuzumab vedotin-piiq	Polivy	Genentech, Roche (Basel, Switzerland)	Relapsed or refractory diffuse large B-cell lymphoma	CD79	MC-Val-Cit-PABC	MMAE	2019
Moxetumomab pasudotox	Lumoxiti	Astrazeneca (Cambridge, UK)	Relapsed or refractory hairy cell leukemia	CD22	MC-Val-Cit-PABC	PE38	2018
Inotuzumab ozogamicin	Besponsa	Pfizer/Wyeth (New York, NY, USA)	B-cell acute lymphocytic leukemia	CD22	Hydrazone	N-acetyl-γ calicheamicin	2017
Trastuzumab emtansine	Kadcyla	Genentech, Roche (Basel, Switzerland)	HER2-positive breast cancer	HER2	SMCC	DM1	2013
Brentuximab vedotin	Adcetris	Seagen Genetics, Millennium/Takeda (Tokyo, Japan)	Anaplastic large-cell lymphoma	CD30	MC-Val-Cit-PABC	MMAE	2011
Gemtuzumab ozogamicin	Mylotarg	Pfizer/Wyeth (New York, NY, USA)	Acute myeloid leukemia	CD33	Hydrazone	N-acetyl-γ calicheamicin	2000; reapproved 2017

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
