# Peer review of "Trends in the Development of Antibody-Drug Conjugates for Cancer Therapy"

_2073-4468, 2023, doi:10.3390/antib12040072_

Round 1
Reviewer 1 Report
Comments and Suggestions for Authors
The authors review some of the history and new developments in the field of antibody drug conjugates. This is still a relatively new field that builds upon the success of monoclonal antibody therapies. Having said that there are some issues with the review that the authors need to address before this manuscript is suitable for publication.
First there seem to be some statements in the introduction that are misleading or incorrect. The sentence starting on line 46 about third generation immuno-oncology drugs seems to imply that all three therapies eliminate cancer cells by disrupting receptors on the cell surface. While this is mainly true for monoclonal antibody therapies immune cell therapies and checkpoint inhibitors stimulate the immune system to eliminate cancer cells. Immune system recognition of cancers cells depends on proteins on the cancer cell surface but these proteins do not necessarily need to be involved in cell signaling. As an example T-cell based therapies recognize cancer cell specific peptides in MHC class 1 receptors on cancer cells. Second the statement about Rituximab is incorrect. Rituximab is an anti-CD20 antibody that eliminates cells by Fc-effector functions. It is not a radioimmuno therapy. Rituximab can be combined with chemotherapy or radiotherapy but it is not a radiotherapy itself. Some monoclonal antibodies have been linked to radioisotopes but this has been mainly for diagnostic purposes only.
Some minor points: When ADC is used as an adverb it should not be plural. For example line 100 "ADCs market" should be "ADC market". Also in my opinion Figure 4 is not necessary. I might be more informative to show some of the chemical structures for ADC linkers in its place.
Comments on the Quality of English LanguageWhen ADC is used as an adverb it should not be plural. For example line 100 "ADCs market" should be "ADC market". This type of error occurs frequently in the manuscript.
Author Response
Response to the reviewers' comments
We thank the editor and reviewers for considering our study and we appreciate the time and care taken to provide us with valuable suggestions to improve the manuscript. We have addressed the specific concerns of the reviewers as indicated below and have revised our manuscript accordingly. We hope that our revised paper now meets the criteria for publication in Antibodies.
Response to the Reviewer #1
The authors review some of the history and new developments in the field of antibody drug conjugates. This is still a relatively new field that builds upon the success of monoclonal antibody therapies. Having said that there are some issues with the review that the authors need to address before this manuscript is suitable for publication.
- First there seem to be some statements in the introduction that are misleading or incorrect. The sentence starting on line 46 about third generation immuno-oncology drugs seems to imply that all three therapies eliminate cancer cells by disrupting receptors on the cell surface. While this is mainly true for monoclonal antibody therapies immune cell therapies and checkpoint inhibitors stimulate the immune system to eliminate cancer cells. Immune system recognition of cancers cells depends on proteins on the cancer cell surface but these proteins do not necessarily need to be involved in cell signaling. As an example T-cell based therapies recognize cancer cell specific peptides in MHC class 1 receptors on cancer cells.
→ We appreciate the reviewer’s suggestions. We have modified the text on line 64-67 as suggested. The changes and additions have been marked in red.
- Second the statement about Rituximab is incorrect. Rituximab is an anti-CD20 antibody that eliminates cells by Fc-effector functions. It is not a radioimmuno therapy. Rituximab can be combined with chemotherapy or radiotherapy but it is not a radiotherapy itself. Some monoclonal antibodies have been linked to radioisotopes but this has been mainly for diagnostic purposes only.
→ In accordance with your suggestion, we have modified the text on line 72-74. The changes and additions have been marked in red.
- Some minor points: When ADC is used as an adverb it should not be plural. For example line 100 "ADCs market" should be "ADC market".
→ In accordance with your suggestion, we have modified the text on line 120, 678. The changes and additions have been marked in red.
- Also in my opinion Figure 4 is not necessary. I might be more informative to show some of the chemical structures for ADC linkers in its place.
→ In accordance with your suggestion, we have removed Figure 4(Homogeneity ADC, Heterogeneity ADC) and replaced it with the chemical structures for ADC linkers found in figure 4.

Reviewer 2 Report
Comments and Suggestions for Authors
Thank you for submitting your review on the latest trends in Antibody-Drug Conjugates (ADCs). I appreciate the depth and breadth of your research, especially concerning clinical trials. I have several suggestions that might enhance the comprehensiveness and accuracy of your manuscript.
- Regarding the Antibody Section:
- It would be beneficial if the manuscript could touch upon the common antigen copy numbers associated with ADCs.
- An important consideration in ADC development is the differential expression between normal cells and target cancer cells. Addressing the difference in expression levels that make an ADC promising for therapeutic applications would be valuable.
- Addressing the major contributors to non-specific uptake originating from the antibody would be beneficial. Discussing which among them – be it Fc function-derived, FcRn-derived, or lectin receptor-derived – has the most significant contribution would be valuable.
- Figure 2: Incorporating information about the 'By-stander effect' might be a significant addition.
- Concerning Conjugation:
- Highlighting the advantages of site-specific conjugation, especially its role in streamlining CMC activities, would be beneficial. For instance, ADCs produced through site-specific methods can ease the development of novel analytical techniques and ensure consistency between batches. In this context, it might be worthwhile to include the reference you provided: https://pubs.acs.org/doi/10.1021/jasms.0c00129
- It's worth noting the emerging trend of utilizing Fc affinity molecules to produce site-specific ADCs without the need for antibody engineering. Please add relevant references.
- Regarding Clinical ADCs: While the manuscript seems to rely on sources like Google Scholar, I suggest cross-referencing with the Beacon database, a more specialized source for ADC clinical trials.
- Payloads: Addressing non-cytotoxic payloads like immunostimulators seems crucial. For instance, Bolt Therapeutics is advancing clinical trials with ADCs using TLR7/8 agonists. Additionally, companies like Takeda and Mersana are exploring STING inhibitors. These developments with ISAC-ADCs represent significant trends, and their absence from this review is conspicuous.
I believe addressing these points will further enhance the quality and comprehensiveness of your review. I look forward to seeing a revised version that incorporates these suggestions.
Author Response
Response to the reviewers' comments
We thank the editor and reviewers for considering our study and we appreciate the time and care taken to provide us with valuable suggestions to improve the manuscript. We have addressed the specific concerns of the reviewers as indicated below and have revised our manuscript accordingly. We hope that our revised paper now meets the criteria for publication in Antibodies.
Response to the Reviewer #2
Thank you for submitting your review on the latest trends in Antibody-Drug Conjugates (ADCs). I appreciate the depth and breadth of your research, especially concerning clinical trials. I have several suggestions that might enhance the comprehensiveness and accuracy of your manuscript.
- Regarding the Antibody Section:
- It would be beneficial if the manuscript could touch upon the common antigen copy numbers associated with ADCs.
→ We appreciate the reviewer’s suggestions. We have modified the text on line 185-190 as suggested. The changes and additions have been marked in red.
- An important consideration in ADC development is the differential expression between normal cells and target cancer cells. Addressing the difference in expression levels that make an ADC promising for therapeutic applications would be valuable.
→ We appreciate the reviewer’s suggestions. We have modified the text on line 194-203 as suggested. The changes and additions have been marked in red.
- Addressing the major contributors to non-specific uptake originating from the antibody would be beneficial. Discussing which among them – be it Fc function-derived, FcRn-derived, or lectin receptor-derived – has the most significant contribution would be valuable.
→ We appreciate the reviewer’s suggestions. We have modified the text on line 214-226 as suggested. The changes and additions have been marked in red.
- Figure 2: Incorporating information about the 'By-stander effect' might be a significant addition.
→ In accordance with your suggestion, we have modified Figure 2.
- Concerning Conjugation:
- Highlighting the advantages of site-specific conjugation, especially its role in streamlining CMC activities, would be beneficial. For instance, ADCs produced through site-specific methods can ease the development of novel analytical techniques and ensure consistency between batches. In this context, it might be worthwhile to include the reference you provided:https://pubs.acs.org/doi/10.1021/jasms.0c00129
→ In accordance with your suggestion, we have modified the text and added reference on line 447-453 as suggested. The changes and additions have been marked in red.
- It's worth noting the emerging trend of utilizing Fc affinity molecules to produce site-specific ADCs without the need for antibody engineering. Please add relevant references.
→ In accordance with your suggestion, we have modified the text and added reference on line 456-464 as suggested. The changes and additions have been marked in red.
- Regarding Clinical ADCs: While the manuscript seems to rely on sources like Google Scholar, I suggest cross-referencing with the Beacon database, a more specialized source for ADC clinical trials.
→ We conducted research on the clinical status of ADCs between 2018 and 2023, investigating clinical.gov, where we identified data from 82 clinical trials. Since the Beacon database you recommended requires a fee, so we opted to explore and 46 more adapt data from EU clinical trials register, Health Canada clinical database, and Google Scholar accessible clinical databases. We've also modified the text, figure, supplementary table to reflect the added data. The changes and additions have been marked in red.
- Payloads: Addressing non-cytotoxic payloads like immunostimulators seems crucial. For instance, Bolt Therapeutics is advancing clinical trials with ADCs using TLR7/8 agonists. Additionally, companies like Takeda and Mersana are exploring STING inhibitors. These developments with ISAC-ADCs represent significant trends, and their absence from this review is conspicuous.
→ In accordance with your suggestion, we have modified the text and added reference on line 353-365 as suggested. The changes and additions have been marked in red.
I believe addressing these points will further enhance the quality and comprehensiveness of your review. I look forward to seeing a revised version that incorporates these suggestions.

Round 2
Reviewer 1 Report
Comments and Suggestions for Authors
The revised manuscript is a more accurate representation of the topic.
Comments on the Quality of English LanguageThere are still instances of double plurals.
Author Response
In accordance with your suggestion, we've changed more than 10 ADCs plurals to singular and marked them in red in the manuscript.
Reviewer 2 Report
Comments and Suggestions for Authors
No additional questions from the reviewer. Great work.
Author Response
Thank you for your valuable suggestions to improve the manuscript.